# Perceptions and experiences of older adults in a youth community health volunteer–led health coaching program in Singapore: A qualitative study

Audrey Shu Ting Kwan[1,2,3☯], Jasmine Yee Ru Cheng[1,2☯], Jed Jasman[1,2],
Elliott Weizhi Sim[1], Alicia Shi Yao Chee[1], Ka Shing Yow[1,4], Jie Xin Lim[1], Si Qi Lim[1,5],
Thaddeus Chi En Cheong[1,6], Nerice Heng Wen Ngiam[1,6], Muhammad Razzan Razaki[1,7],
Xiaoting Huang[1,8], Lynn Pei Zhen Teo[1,2], Sheryl Wen Ning Ho[1,2], Zi Hao Lee[1,2],
Kharuna Jaichandra[1,2], Chee Hsiang Liow[9], Natasha Howard[9,10‡],
Kennedy Yao Yi Ng[1,2,11,12,13*‡], Lian Leng Low[2,14,15‡]

1 TriGen, Singapore, Singapore, 2 Division of Population Health and Integrated Care, Singapore General Hospital, Singapore, Singapore, 3 Department of Physiotherapy, Singapore General Hospital, Singapore, Singapore, 4 Department of Internal Medicine, National University Health System, Singapore, Singapore, 5 Department of Physiotherapy, Singapore General Hospital, Singapore, Singapore, 6 Department of Internal Medicine, Singapore General Hospital, Singapore, Singapore, 7 Community Hospital Referral Team, SingHealth Community Hospital, Singapore, Singapore, 8 Department of Geriatric Medicine, Singapore General Hospital, Singapore, Singapore, 9 Saw Swee Hock School of Public Health, National University of Singapore, Singapore, Singapore, 10 Faculty of Public Health and Policy, London School of Hygiene and Tropical Medicine, London, United Kingdom, 11 Division of Medical Oncology, National Cancer Centre, Singapore, Singapore, 12 Oncology Academic Clinical Programmes, Duke-NUS Medical School, Singapore, Singapore, 13 Department of Global Health and Population, Harvard T.H Chan School of Public Health, Boston, United States of America, 14 Research and Translation Innovation Office, SingHealth Community Hospitals, Singapore, Singapore, 15 Centre of Population Health and Implementation Research, SingHealth Regional Health Services, Singapore

☯ These authors contributed equally to this work.
‡ These authors also contributed equally to this work
* kennedy.ng.y.y@singhealth.com.sg

## Abstract

### Background

With an aging global population, there is a growing emphasis on community-based approaches to preventive health for older adults. While Community Health Volunteers have been found to be effective in empowering older adults to manage their chronic conditions, there is limited research on the role of Youth Community Health Volunteers (YCHVs) and little is known about how older adults perceive their role and effectiveness.

### Objectives

This qualitative study explores older adults' experiences and perceptions of a YCHV-led health coaching program following community health screening in Singapore.

**Data availability statement:** The data supporting this study are available from the authors upon request. The de-identified dataset generated and analyzed during this study is held securely by the study team at Singapore General Hospital. Data access requests may be directed to the Singapore General Hospital Research & Innovation Office (rio@sgh.com. sg), which serves as the institutional point of contact and will facilitate requests to ensure long-term data availability.

**Funding:** The study was funded by Singapore's National Medical Research Council under the SingHealth Regional Health System PULSES Centre Grant (NMRC/CG/C027/2017_SHS), the Healthy, Empowered and Active Living (HEAL) Fund, the Infocomm Media Development Authority Digital for Life Fund, Singapore Ministry of Health's National Medical Research Council Population Health Research Grant (PHRGTC-7-0001), and SingHealth Academic Medicine HEART grant (AM/HRT025/2023). The funders had no role in study design, data collection and analysis, decision to publish, or preparation of the manuscript.

**Competing interests:** The authors have declared that no competing interests exist.

## Methods

We conducted one-to-one semi-structured interviews with 19 older adults who had participated in HealthStart, a YCHV-led health coaching program. Purposive sampling was used to ensure representation across diverse demographic backgrounds and varying levels of program engagement. Interviews were carried out between November 2023 and January 2024 in community centers, healthcare facilities, or via telephone/Zoom, and explored participants' experiences with the program, perceptions of YCHVs, and health-related outcomes. We analyzed data thematically using deductive and inductive coding.

## Results

We generated five themes: (1) How YCHV facilitation contributed to prevention outcomes, (2) YCHV facilitation techniques, (3) Intrinsic factors influencing older adults' receptivity towards YCHVs, (4) Older adults' perceptions of YCHVs, and (5) YCHVs' potential to be further empowered as health advocates. Findings provided insights into health behaviors of older adults, the perceived role of YCHVs in influencing these behaviors, and potential and opportunities to strengthen YCHVs' capacity as health advocates.

## Conclusion

Engaging YCHVs as preventive health advocates for older adults appeared both acceptable and promising. This study highlights the potential for YCHV scale-up, while emphasizing the need for targeted training in behavior change and referral to community resources. Beyond individual health gains, such initiatives may help address ageist attitudes, foster intergenerational solidarity, and complement primary care efforts.

## Introduction

Over the past few decades, the global population has undergone a significant demographic shift, with older adults making up an increasing proportion of society [1]. By 2050, the number of people aged 60 and above is expected to more than double from 2015, reaching 2.1 billion [2]. This aging trend has led to a rise in chronic diseases [2], which pose substantial challenges at both the individual and systemic levels. Older adults with chronic conditions often face persistent symptoms, functional decline, higher healthcare costs, and reduced quality of life [3]. The growing burden of chronic diseases also places a strain on the healthcare system, resulting in increasing demand for medical services, higher hospital bed occupancy, and emergency department visits [4]. Given these challenges, there is a growing emphasis on empowering community-dwelling older adults to actively manage their health.

Community-based interventions have emerged as a key strategy in supporting older adults, with community health volunteers (CHVs) playing a critical role. CHVs,

individuals from the community trained as health advocates, have gained global recognition for their ability to bridge healthcare systems and community needs [5]. For instance, programs such as the Coaching Ongoing Momentum Building On Stroke Recovery Journey (COMBO-KEY) initiative have demonstrated the effectiveness of trained volunteers in improving stroke survivors' self-efficacy and engagement in self-management behaviors [6]. The Health Teams Advancing Patient Experience: Strengthening Quality (Health TAPESTRY) program similarly leveraged trained volunteers to help older adults set lifestyle goals and connect with community resources [7], leading to fewer hospital admissions and increased physical activity [8].

Despite growing evidence on the impact of CHVs, research on the role of youth as community health volunteers (YCHVs) remains limited. While youth involvement in aging and intergenerational programs has been documented extensively, existing initiatives are often broad in scope, and few focus specifically on health promotion. To our knowledge, no studies have examined the role, acceptability, and potential of YCHVs from the perspective of older adult beneficiaries. Negative ageist attitudes toward youth may also pose challenges to scaling YCHV-led programs. Understanding their experiences and perceptions can offer valuable insights into the feasibility and impact of YCHV-led health interventions.

We thus aimed to explore the experiences and perspectives of older adults who participated in a health coaching program delivered by YCHVs, contributing to broader discourses on innovative, community-driven approaches to preventive health.

## Methods

### Study design

We employed a qualitative study design, interviewing older adults participating in HealthStart, a YCHV-led health advocacy program in Singapore.

### Study site

This study contributed to an evaluation of the HealthStart program, jointly organized by TriGen Ltd [9], a charity organization, and the Singapore General Hospital (SGH) Division of Population Health and Integrated Care (PHIC). The program was embedded in Singapore's public healthcare system, where government subsidies reduce out-of-pocket costs for primary and specialist care. Universal catastrophic coverage is provided through MediShield Life, Singapore's national health insurance scheme, complemented by compulsory medical savings (MediSave) and a safety net fund (MediFund) [10]. This mixed financing model differs from countries without government-sponsored coverage, where affordability may be a significant barrier to follow-up.

HealthStart aims to increase engagement and primary care follow-up rates among older adults attending community health screenings for chronic cardiovascular disease by training and empowering YCHVs as health advocates. YCHVs, aged 15–35 years, were recruited from schools, volunteer portals, and youth networks, with no prior healthcare training required. All YCHVs completed a blended training program with online modules and a one-day workshop on chronic disease management, SMART (Specific, Measurable, Achievable, Relevant, and Time-bound) goal setting, motivational interviewing, and the Singapore healthcare system. Competency was assessed through quizzes and role-plays, and only those who passed were deployed. Each HealthStart team comprised one healthcare volunteer (doctor, nurse, or allied health professional) and four YCHVs, with each YCHV typically supporting two older adults. Across two program cycles, 149 YCHVs were recruited, 138 trained, and 102 deployed (mean age 24 years; 74% female, mostly post-secondary students). YCHVs followed participants for three months with at least six in-person or virtual sessions, helping them understand their conditions, enroll with a primary care provider, use a health app (HealthHub or Healthy365), and set personalized SMART lifestyle goals under professional supervision. Further details about the HealthStart program [11] and the mixed-methods outcome evaluation [12] can be found in the cited publications.

## Sampling and recruitment

We used heterogeneous purposive sampling to provide some representation across different demographics (i.e., age, gender, race, education level, number of follow-ups within the program, and post-program primary care follow-up status). Eligible participants were contacted by phone or text messages after screening and invited to participate. A few individuals declined participation, though the exact number was not systematically recorded. Recruitment continued until data saturation was reached, determined collectively by researchers (ASTK, JYRC, JJ, EWS) when further interviews yielded no new insights.

All participants received information sheets, had opportunities to ask questions, were regularly informed they could decline to answer any question or withdraw at any point, and provided written informed consent prior to participation – including audio-recording and use of anonymized quotes. All data were de-identified and stored on electronic devices only accessible to the research team.

## Data collection

We developed the interview guide based on program Theory of Change (Fig 1), which included perceptions of the program, YCHVs, and health outcomes (S1 Table).

Data was collected through one-to-one, semi-structured interviews with older adults conducted between November 2023 and January 2024. Interviews were conducted in a quiet room in community centers, healthcare facilities, via telephone, or Zoom (Zoom Video Communications, Inc., San Jose, California) by four co-authors (ASTK, JYRC, JJ, EWS), all affiliated with the Division of Population Health and Integrated Care (PHIC) at Singapore General Hospital. The

**Legend**
Self Determination Theory: →
S.M.A.R.T. Goals: →
Intergenerational & Service Learning: →
HCV = Healthcare Volunteer
YCHV = Youth Community Health Volunteer

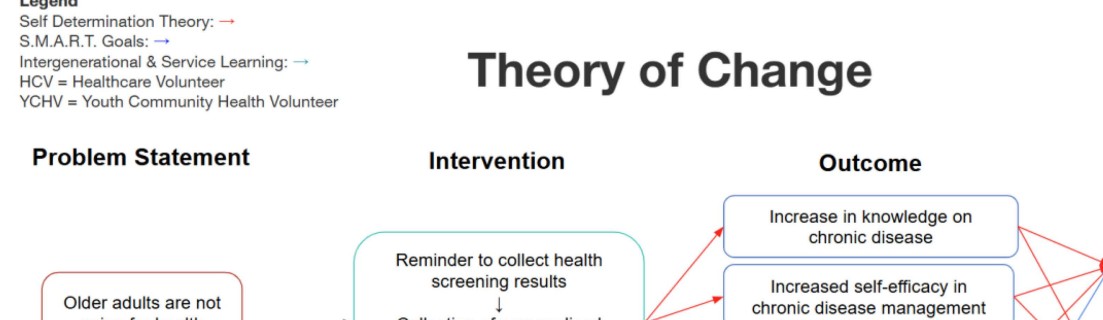
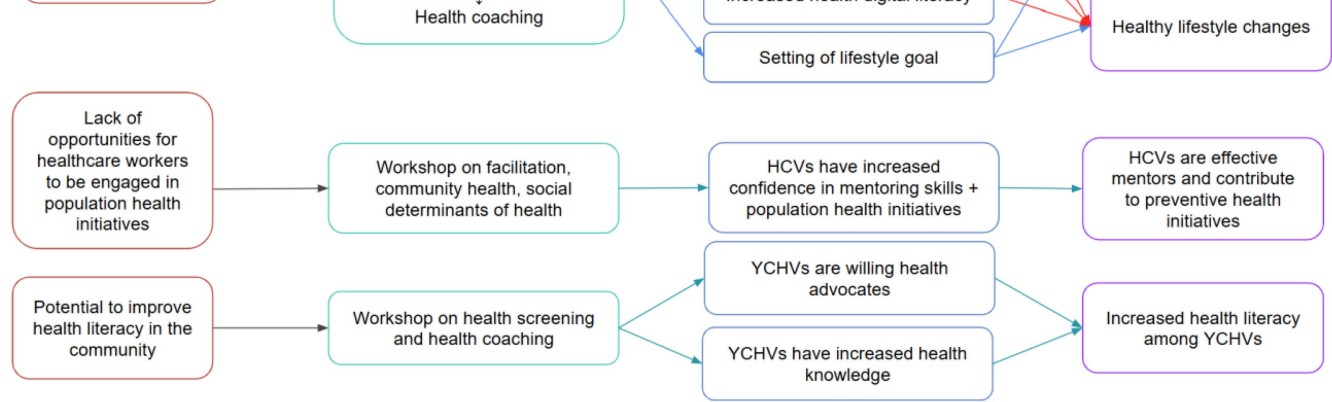

**Fig 1. HealthStart Program Theory of Change.**

interviewers were public health researchers or researchers-in-training, relatively new to qualitative research, and received guidance on the use of the interview guide to ensure consistency across interviews. Interviews were supervised by the qualitative lead (ASTK). Interviews were conducted in English or Mandarin, depending on participant preference. For Mandarin interviews, audio recordings were translated directly into English transcripts by bilingual co-authors fluent in both languages to ensure accuracy and consistency. Only the interviewer and participant were present during each interview. Each interview lasted approximately 60 minutes, was scheduled at participants' convenience, audio-recorded, and accompanied by field notes documenting contextual observations. Transcripts were generated using automated transcription software, manually checked for verbatim accuracy, and identifiable information redacted. Transcripts were not returned to participants for comment or correction.

### Data analysis

Data were analysed thematically, as described by Braun and Clarke [13], employing QSR NVivo 14 Software for data management and analysis. Four team members (ASTK, JYRC, JJ, EWS) coded transcripts using a deductive-abductive approach, focusing on program perceptions and health outcomes, while remaining open to unexpected themes. After independent coding, a secondary coder reviewed them to finalize and address discrepancies. The codes were then organized into broader themes and the research team developed and reviewed the themes through an iterative process. To ensure intercoder reliability, any disagreement was resolved through a discussion among team members.

### Reflexivity

Most authors are public health researchers or researchers-in-training with interest in community health and affiliated with SGH's Division of Population Health and Integrated Care. While some team members (ASTK, JYRC, JJ, EWS) previously assisted in HealthStart activities, they had minimal contact with any participants before the study. Participants were informed that interviewers were researchers from SGH and TriGen evaluating the HealthStart program. The authors recognize that direct translation from Mandarin to English may have introduced minor loss of nuance, which they mitigated through cross-checking by bilingual researchers. This study is reported in line with the Consolidated Criteria for Reporting Qualitative Research (COREQ) checklist (S2 Table) [14].

### Ethics

This study received approval from the SingHealth Centralised Institutional Review Board (reference 2023/2480).

## Results

### Participant characteristics

Table 1 shows our 19 participants, most were aged 60–69 (68%; n = 13), male (58%; n = 11), Chinese (79%; n = 15), married (47%; n = 9), owned their residence (53%; n = 10), and completed final primary care follow-up (74%; n = 14). Education (i.e., 32% completing primary, 26% secondary, 37% tertiary education) and number of YCHV engagements (i.e., 21% completing 1 engagement, 32% 2–3 engagements, 26% 4–5 engagements, 21% 6–7 engagements) were more evenly spread.

### Analytical themes

We identified five themes: (1) How YCHV facilitation contributed to prevention outcomes, (2) YCHV facilitation techniques, (3) Intrinsic factors influencing older adults' receptivity towards YCHVs, (4) Older adults' perceptions of YCHVs, and (5) YCHVs' potential to be further empowered as health advocates.

**Table 1. Participant characteristics.**

| ID | Age group (years) | Gender | Ethnicity | Marital Status | Education Level | Residence | No. of YCHV Engagements | Final Primary Care Follow-up |
|---|---|---|---|---|---|---|---|---|
| OA01 | 60-69 | Male | Chinese | Divorced | Tertiary | Rent | 3 | Yes |
| OA02 | 60-69 | Male | Chinese | Single | Primary | Own | 4 | Yes |
| OA03 | 80-89 | Male | Chinese | Married | Tertiary | Own | 2 | Yes |
| OA04 | 80-89 | Male | Chinese | Single | Secondary | Own | 3 | No |
| OA05 | 60-69 | Female | Chinese | Widowed | Primary | Rent | 4 | Yes |
| OA06 | 60-69 | Male | Malay | Married | Primary | Rent | 3 | Yes |
| OA07 | 50-59 | Female | Chinese | Married | Tertiary | Own | 7 | Yes |
| OA08 | 60-69 | Female | Chinese | Married | Tertiary | Own | 5 | Yes |
| OA09 | 70-79 | Male | Chinese | Married | Secondary | Own | 4 | Yes |
| OA10 | 60-69 | Male | Chinese | Divorced | Secondary | Lodge | 6 | Yes |
| OA11 | 60-69 | Male | Chinese | Married | Primary | Rent | 6 | No |
| OA12 | 50-59 | Female | Chinese | Married | NIL | Rent | 6 | No |
| OA13 | 60-69 | Female | Indian | Married | Primary | Own | 2 | Yes |
| OA14 | 50-59 | Male | Malay | Married | Tertiary | Rent | 3 | No |
| OA15 | 60-69 | Female | Chinese | Single | Tertiary | Own | 4 | Yes |
| OA16 | 60-69 | Female | Chinese | Single | Secondary | Own | 1 | Yes |
| OA17 | 60-69 | Female | Indian | Widowed | Secondary | Rent | 1 | Yes |
| OA18 | 60-69 | Male | Chinese | Single | Primary | Rent | 1 | No |
| OA19 | 60-69 | Male | Chinese | Single | Tertiary | Own | 1 | Yes |

### Theme 1: How YCHV facilitation contributed to prevention outcomes

Four sub-themes: (i) increased health knowledge; (ii) nudged primary care follow-up; (iii) lifestyle modification and self-efficacy; and (iv) improved health-related digital skills, revealed how older adults' knowledge, attitudes, behaviors, and physical well-being changed through YCHV engagements.

**Sub-theme 1.1: Increased health knowledge.** Many older adults reported gaining valuable health knowledge through their interactions with YCHVs. Participants described developing a clearer understanding of chronic conditions such as hypertension, hyperlipidemia, and diabetes, as well as the key health behaviors needed to manage these conditions. These included consuming nutritious foods like vegetables and whole grains, reducing the intake of sugary, salty, or oily foods, controlling portion sizes, and recognizing the importance of regular physical activity. Some participants described the information shared by YCHVs as entirely new, while others appreciated clarifications about the severity of their health conditions or misconceptions and knowledge gaps. For instance, one shared how her misunderstanding of hyperlipidemia was corrected:

*"Yeah [the YCHV explanation was] new information because I was thinking, huh, I'm not like fat or anything. [YCHV] said, it's not necessary to be plump or fat to have cholesterol. Your stature doesn't determine, you know, whether you have cholesterol or not… so now I understand." (OA17)*

Another participant highlighted how he had not initially appreciated the seriousness of his newly diagnosed condition until YCHVs explained its implications:

*Interviewer03: For you, before [YCHV] came—because you have high cholesterol and such… did you feel that it was serious?*

*OA05: No ah… [But] the volunteers said "even though it is not that high yet but this is only the start, and you still don't pay attention", so I listened...*

**Sub-theme 1.2: Nudged primary care follow-up.** Some older adults reported that they had already been attending regular follow-ups with primary care providers, such as General Practitioners or Polyclinics (subsidized, one-stop primary healthcare centers) [10], to monitor their chronic conditions even before the program. One participant shared that he was under the care of a specialist at a tertiary hospital for ongoing management and did not require additional encouragement from YCHVs.

For those who had abnormal health screening results but were not yet consulting a primary healthcare provider, YCHVs encouraged them to initiate appointments and share their screening outcomes with their doctors. Some participants acknowledged the importance of this advice and subsequently sought necessary medical attention, as illustrated by one participant:

*"[YCHV] asked me to see doctor. Then I go to poly[clinic], then I go check and take the medicine, cholesterol medicine, I take. Before that I never take." (OA13)*

However, a few did not follow up with a primary care provider by the end of the program and indicated they were confident managing their health independently. They often described their conditions as minor or borderline, not requiring medical attention:

*"Oh, no, no, no. I said, I said so far okay, my energy is good. I rarely see a doctor, the volunteers also see me healthy, because I prioritize my health, and then I have no problem. I usually listen to the radio, then do some house chores and use it as exercise." (OA04)*

**Sub-theme 1.3: Lifestyle modification and self-efficacy.** Many older adults reported engaging in increased physical activity and adopting various forms of exercise. These ranged from tracking higher step counts and dancing at home to utilizing neighborhood exercise corners and gardens, following workout videos on social media platforms, and participating in community exercise programs.

Many also mentioned dietary changes, such as reducing their consumption of sugar-sweetened beverages, processed foods, fast food, and fried foods, including certain local Indian and Chinese dishes. A few interviewees altered their food preparation methods, opting for steaming and blanching, using less salt, and reducing portion sizes. Increased intake of fruits, vegetables, and whole grains such as oats and brown rice was also reported. Some participants noted health improvements, including increased energy levels, weight loss, and improved cholesterol levels, as evidenced during subsequent medical reviews.

One participant shared how she not only modified her lifestyle with the YCHV's guidance but also encouraged her family to adopt similar habits:

*"[YCHV] is actually very good. She just said, need to exercise. At that time, I was quite fat, quite heavy, and she said to cut down on eating these [unhealthy] things. I listened to her… and lost a few kilograms... Previously, I would go and buy take-out for breakfast, and they were oily. Then, after I listened to her, I started eating oatmeal every day... whenever I cook dishes, I don't use oil. Keep making soup… and pair it with rice. For vegetables... I use blanching to cook… And then, afterward, I told my daughter—I live with my daughter—'You need to eat these with me.'"* (OA05)

A small number of participants demonstrated self-efficacy by independently monitoring their blood pressure and managing their health. Many older adults were able to sustain these routines and healthy habits beyond the program's conclusion.

**Sub-theme 1.4: Health-related digital skills.** While learning a digital health application was a secondary aim of the program, many participants already had health applications installed on their mobile phones prior to their involvement. The most frequently cited use was booking appointments with healthcare providers. A few participants utilized these applications to track their step counts and accumulate health-related "points," which could be redeemed for incentives. Despite these uses, concerns about online scams were noted as a barrier to fully engaging with such applications. Additionally, participants with lower health and digital literacy faced challenges in understanding or utilizing the full functionality of these tools. YCHVs played a crucial role in addressing these barriers by providing guidance and encouragement. Beyond health-related apps, they also helped older adults develop broader digital literacy skills, making them more confident in navigating technology. For example, one beneficiary couple, initially unfamiliar with WhatsApp, learned to use its voice messaging feature with guidance from their YCHV. This skill became their primary mode of communication with the YCHV during and after the program.

**Theme 2: YCHV facilitation techniques**

This theme highlighted the techniques YCHVs employed to foster trust and accountability, enabling older adults to achieve their health goals. Our 2 sub-themes were: (i) personalized education and goal-setting; and (ii) relationship building, motivating, and nudging.

**Sub-theme 2.1: Personalized education and goal-setting.** Older adults appreciated how YCHVs explained their health report findings in a clear and personalized manner, tailoring health advice to suit diverse backgrounds, languages, and lifestyles. Health-related lifestyle goals commonly focused on improving diet and increasing physical activity, with small, achievable targets customized to each individual's circumstances. One participant described how the YCHV collaborated with her to set a goal of increasing physical activity, gamifying the process using a national health app:

*"…the 365 app (national health phone application), we always want that points, health points and then want to convert the points to redeem the points and then [convert] for the ezlink card (public transport card). So, so you want this you have to walk a bit lah… Make sure that MVPA (moderate to vigorous physical activity) hits 30 minutes per day and above 5000 (steps)."* (OA08)

**Sub-theme 2.2: Relationship building, motivating, and nudging.** Participants shared that they felt cared for by the YCHVs, particularly through regular check-ins about their health and progress with lifestyle goals via text messages or calls. One continued to regularly connect with her YCHV through WhatsApp voice messages. For some, these interactions felt like casual conversations, creating a comfortable environment where they felt at ease to ask questions:

*"[YCHV] would call to ask how am I, et cetera, and do I need help et cetera. She did a great job! … she would care and then she would listen to what I have to say and then when I asked questions (about my health), she would- provide me with feedback and suggestions… like having chit-chats with friends…" (OA05)*

These regular check-ins also helped participants stay accountable for their health goals, as YCHVs encouraged and motivated them to continue engaging in healthy habits. One participant, whose husband also joined the program, appreciated that these nudges came from someone outside of their family:

*"Sometimes need [a] third party to talk [to the older adults], then [they] will listen ah. [For] some people, when [their] wives or husbands talk, [they] don't want to listen, [but] when third party talks ah, then they listen." (OA06)*

**Theme 3: Intrinsic factors influencing older adults' receptivity towards YCHVs**

This theme revealed the personal beliefs, priorities, and motivations that shaped older adults' engagement with YCHVs' health coaching. Our two sub-themes were: (i) perceived challenges and limitations; and (ii) personal motivations and ownership.

**Sub-theme 3.1: Perceived challenges and limitations.** Several barriers affected older adults' receptivity to YCHVs' health coaching and lifestyle advice. Some participants viewed health issues as an inevitable consequence of aging, leading to a sense of resignation. One participant, for instance, dismissed the seriousness of his health results, believing it was a natural part of growing older:

> *Interviewer03: Understand, understand. Do you feel that the results were serious?*
>
> *R09: It was okay leh… When you're older, all these blood pressure and blood sugar would be high. For instance, my blood pressure is around 130 or 140.*

Others perceived their conditions as mild and not requiring follow-up with a doctor.

Many cited practical challenges such as busy schedules, caregiving responsibilities, and competing priorities. For example, one older adult struggling with unemployment described how his immediate concerns about physical pain and livelihood overshadowed lifestyle changes:

> *"If I can't work, I don't have income mah. I have to go for doctor's appointment, I still need to, [look out] for myself... Then I still have my mother whom I have to care for ah. Eighty plus years old... Her blood pressure when she measures, [the readings are] 170+, 180+, 160+... I am in pain all over, I don't… I don't have the…mood lah..." (OA18)*

Cultural and environmental factors also played a role. One participant highlighted the difficulty of maintaining a healthy diet due to the cost and availability of nutritious options:

> *"The European so-called, their diet is very healthy compared to Singapore local [diet]... Even though you might get [healthy food] from NTUC (large local supermarket chain) the cheapest, five dollars around… I find that's still not enough because I myself will feel [hungry]. Because if you go out to the hawker center (an open-air food court offering affordable local dishes), it's very difficult [to eat healthily]… Singapore local food ah, …is very big portion.... Like one bowl of fishball noodle is same price ah or even cheaper than the packet of uh- salad from the NTUC… I really wish to work out something with the uh NTUC, our government supermarket… maybe each hawker center ah, we have a stall there, so is provide all healthy food." (OA16)*

Barriers to consulting primary care physicians included perceptions of high costs and concerns about over-reliance on Western medicine. Two participants expressed a preference for Traditional Chinese Medicine (TCM) and natural lifestyle changes over long-term medication, as one shared:

> *"They ask me want to take the medicine. I say not lah, cause I heard the moment you start only ah, you must do it forever already, [and] I'm a person that don't like to eat Western medicine… So I may go TCM to check for those, those sign lah like diet…" (OA07)*

Finally, some participants felt they had already implemented and sustained healthy lifestyle changes independently, rendering further YCHV involvement unnecessary. A few had gained health information from other sources, such as the radio, and declined further follow-ups after one or two sessions.

**Sub-theme 3.2: Personal motivations and ownership.** For many participants, a sense of personal responsibility and motivation played a pivotal role in adopting healthier behaviors. Family was a key driver—participants wanted to avoid becoming a burden to their children or spouse, continue caregiving roles, or witness their grandchildren's milestones. One participant shared his motivation:

*"I do not want to be a burden to my children… I do not want my children to spend money on me… the one favor that I can do for my two children ah… you have to take good care of your health lah. I just want to lead a healthy life… I would like go see the world, one day maybe five years from now." (OA01)*

Others were motivated by a desire to minimize dependence on medication. Participants who embraced healthier behaviors often exhibited a readiness to change and believed in taking ownership of their health. While YCHVs provided valuable guidance, participants recognized that sustained change ultimately depended on their own commitment:

*"[The program] did help to remind those patients, like us who went for the health screening [of healthier lifestyle habits]. Whether it did help or not, it is quite dependent on the individual." (OA10)*

Similarly, another older adult reflected that

*"It all depends on whether you want to change… After you were informed, whether you want to do, it is all dependent on you. The (YCHV) shared with me, and I also tried to persuade my friends, but they didn't listen. They would comment, "it's okay even if I die, why do I need to live long?"" (OA05)*

## Theme 4: Older adults' perceptions of YCHVs

This theme revealed the impressions formed by older adults about YCHVs. Our two sub-themes were: (i) relationships brought fresh perspectives and insights on health and lifestyle; and (ii) challenges in engagement and rapport building.

**Sub-theme 4.1: Relationships brought fresh perspectives and insights on health and lifestyle.** Many older adults formed positive impressions of the YCHVs, appreciating their enthusiasm, knowledge, and commitment. Several participants expressed a preference for engaging with younger volunteers, valuing the fresh perspectives and insights they brought to health and lifestyle discussions. Some acknowledged that YCHVs' digital savviness and exposure to contemporary trends enriched the interactions, offering opportunities for learning.

One participant gained new insights when the youths shared new and practical steps to support healthy behavior:

*"...the experience I can get from youngsters is … the new things... because we need input then we can explore. Ah, so it's basically for health and then back to the [importance of] scheduling, timetable. I was- I never think about- of that, you know?... younger [volunteers are] better... youngsters nowadays, they have a lot, you know, they know a lot." (OA14)*

Participants also noted that the age gap did not hinder YCHVs from engaging meaningfully on mature topics. Many praised the YCHVs' passion and their genuine desire to serve, contrasting their efforts with other volunteers they perceived as less committed:

*"...y'all really want to be a volunteer that's why y'all are very good... You can feel who's doing with enthusiasm or not... when people like you [come] to my house, I really feel that it is an extra mile… So, I was very impressed that the young people these days are doing so well, I really think that Singapore really has hope, these younger generations coming this way." (OA15)*

**Sub-theme 4.2: Challenges in engagement and rapport building.** Despite favorable impressions, some older adults noted challenges in engaging with YCHVs, often rooted in generational and experiential gaps. A recurring concern was the perceived disconnect due to differences in life stages, health conditions, and lived experiences, which limited the depth of conversations.

Some participants questioned the YCHVs' credibility, expressing reservations about their ability to address complex health issues due to their lack of professional training. Language barriers exacerbated these challenges, particularly for those more comfortable communicating in languages other than English:

> *"...Language barrier is a problem... if he/she is young, and the age gap is like thirty to forty years, it's very hard for them to understand our concerns... It's just not convincing lah, not like the polyclinic doctor lah… The polyclinic doctors have the 'power'... As for the youth, they are mostly like textbooks, they just regurgitate things from textbooks." (OA19)*

**Theme 5: YCHVs' potential to be further empowered as health advocates**

Two sub-themes: (i) Older adults value and express interest for more of such programs, and (ii) Training YCHVs in behavioral change and mobilizing support systems for older adults, revealed the opportunities to further empower YCHVs as effective health advocates.

**Sub-theme 5.1: Older adults value and express interest for more of such programs.** Many participants expressed appreciation for the program, emphasizing several benefits, including the detailed explanations of health reports provided by YCHVs, the health knowledge gained, and the encouragement to adopt healthier behaviors. The program's role in supporting recipients to set and achieve health goals, the provision of a free program, and the relationships fostered between YCHVs and older adults were particularly valued. Some interviewees also highlighted the potential for such programs to alleviate loneliness among seniors by offering companionship.

A few participants suggested that extending the program's duration or introducing additional initiatives could increase its reach and impact, enabling more older adults to benefit. One participant, for example, appreciated the guidance received from her YCHV in understanding and following up on her health goals—a support she noted was not available during her interactions with her doctor due to time constraints:

> *"[YCHV told me] there was a set of goals after I join HealthierSG (a national program linking residents to a primary care doctor and supporting them to set healthy lifestyle goals). I say, "where's my goals? I got no goals", then she say "oh yah, you look at the HealthHub (local health app)",... So, there's actually goals set down there, but … maybe the doctor don't have so much time, because there's so many patients waiting… so they assume that we know how to go and see. But without (YCHV) coming in, nobody know that there's a goal down there, you know." (OA15)*

**Sub-theme 5.2: Training YCHVs in behavioral change and mobilizing support systems for older adults.** Participants acknowledged the efforts of YCHVs in providing health information and behavioral nudges while emphasizing that behavioral change ultimately depended on the individual's willingness to take ownership. Some participants offered suggestions to strengthen the program, particularly through equipping YCHVs with skills to address resistance among older adults. For instance, one participant underscored the importance of training YCHVs in understanding older adult psychology to facilitate behavioral change:

> *"Yah yah yah, [older adults] have to initiate the change themselves. If they don't change, there is nothing we can do for them... If even I cannot change my parents' [mindset], expecting the youths to be able to change an elderly's mindset is [a challenge]. In fact, the volunteers should be taught how to deal with the challenge [when older adults are resistant to change]. Actually, this part of training is even more needed." (OA19)*

In addition, participants highlighted the importance of mobilizing older adults' existing support networks, such as friends and family, to reinforce their health journeys:

> "I mean, to talk to old people is a bit difficult because they have lived such a long life ah… and they will not listen to people younger. … They are stubborn… So if [a health scare happens to their] surrounding friends ah, they saw it then they will feel scared then they want to do something to their health." (OA07)

While some older adults were aware of community exercise programs or neighborhood activities, many had limited experience participating in them. When asked about exploring these programs with YCHVs, several participants expressed openness, recognizing the potential social benefits.

## Discussion

To our knowledge, this is the first study to explore the role of YCHVs as health advocates for older adults, addressing a critical gap in the literature and practice. Our findings provide novel and practical insights that can inform policy and interventions aimed at addressing the preventive health needs of older adults in the community. A key strength of this study is its inclusion of participants with different demographic backgrounds, including older adults who did not achieve a follow-up with a primary care provider- a primary outcome of the program. The use of semi-structured in-depth interviews allowed for a comprehensive exploration of key themes, capturing perspectives beyond the scope of the interview guide.

Older adults reported useful health gains under YCHV guidance, including increased health knowledge, improved primary care follow-ups, and healthier lifestyle adoption. These findings align with a systematic review by Zhong et al showing intergenerational activities positively impacted older adults' physical function, psychosocial and cognitive health, social relationships, and quality of life [15]. However, most existing programs focused on general befriending or structured group activities, with limited health-specific interventions led by youth. This study demonstrates the potential for youth to facilitate meaningful health conversations and behavior changes among older adults beyond structured exercise sessions. Research by Tadai and Tan [16], and Soundararajan et al [17], in Singapore highlighted that low-income older adults face barriers to adopting digital tools and often lack social and digital support. Given their tech fluency, YCHVs were well-positioned to bridge this gap, providing digital literacy support and safeguarding older adults against online scams [18,19]. This extends YCHVs' role beyond direct health advocacy to include digital empowerment, which is increasingly crucial for navigating modern healthcare systems.

YCHVs employed personalized education, goal-setting, relationship-building, and regular communication and accountability to support older adults in their health journeys. These approaches aligned with the principles of health coaching described by Newman and McDowell [20], which emphasize client-driven goal-setting, motivational communication, and behavior change facilitation. Wolever et al.'s systematic review further supports this by defining health coaching as a patient-centered process grounded in behavioral change theories, delivered by trained individuals who use motivational techniques to empower clients [21]. Despite not being health professionals, the older adults' feedback indicated that YCHVs effectively facilitated behavior change. This suggests that with adequate training, YCHVs can apply core health coaching principles, reinforcing the importance of equipping them with knowledge on behavioral change theories, effective communication strategies, and health navigation skills to maximize their impact.

Intrinsic factors shaped older adults' receptivity to YCHV guidance. Barriers included resignation to aging, competing priorities, limited access to nutritious foods, skepticism toward Western medicine, and a perceived lack of need for external support. Facilitators included strong personal motivation, family obligations, and a sense of ownership over their health. These findings align with a Singapore-based qualitative study by Subramania et al., [22] which identified willpower, discipline, and cultural factors such as traditional diets and festive indulgences, as key influences on lifestyle adoption. Similarly, Chang et al. found that older generations often favored Traditional Chinese Medicine over Western Medicine,

responding passively to younger family members' health suggestions [23]. A study in Canada further emphasized that older adults' lifestyle changes depended on their perceptions of aging, motivation, and self-efficacy [24]. Those who viewed health conditions as inevitable aspects of aging were less likely to engage in preventive behaviors. This underscores the importance of training YCHVs in behavioral change models such as the Theory of Planned Behavior and Self-Determination Theory [25], and equip them with knowledge about complementary medicine to better support older adults in overcoming these barriers.

Older adults held mixed perceptions of YCHVs. Positive attributes included enthusiasm, fresh perspectives, knowledge, and commitment. However, concerns arose regarding language barriers, generational disconnect, and perceived limitations in credibility due to YCHVs' lack of professional training. Some concerns were age-specific, while others related to their role as CHVs. A study on CHVs in Kenya found that community stakeholders perceived them as lacking adequate health knowledge and resources [26]. Similarly, an English study on lay volunteers noted that while they were distinct from healthcare professionals, they were valued for their personalized attention and deep community connections [27]. These findings highlight the need for clear role delineation and adequate training for YCHVs to build credibility while emphasizing their complementary role to healthcare professionals. Structured mentorship and guidance from experienced healthcare volunteers may also strengthen their perceived credibility. Age-related challenges were evident in an evaluation of an intergenerational program in Singapore [28]. Youth participants who were not proficient in Mandarin or dialects struggled to engage in meaningful conversations with older adults, mirroring the language barriers identified in this study. Additionally, some older adults hesitated to express themselves, citing a lack of shared topics of interest. Sun et al. suggested that negative intergenerational stereotypes perpetuate individual-level biases and hinder societal-level age-friendly environments [29]. Structured facilitation of intergenerational interactions has been shown to mitigate these stereotypes [30], reinforcing the broader societal value of YCHV programs beyond immediate health benefits.

Overall, older adults appreciated the YCHV program, citing positive relationships and tangible health benefits. CHVs have been recognized for their ability to address social determinants of health, connect patients to resources, and enhance primary care engagement [31]. Similarly, lay health volunteers in primary care teams have been found to improve patient understanding of community resources and foster stronger healthcare engagement [32]. A distinctive feature of YCHVs is the smaller power differential with older adults, fostering trust and relationships that encourage open communication and support self-motivated behavioral changes. These findings highlight the need for structured training in behavioral change approaches, effective intergenerational communication strategies and navigation of healthcare and community resources. Furthermore, formal integration of YCHVs into primary care and community health systems, with appropriate honorariums for volunteers, could enhance their impact, adding a human touch to healthcare interactions while deepening healthcare providers' understanding of patient needs [33]. By equipping YCHVs with the right skills and positioning them as a valuable component of the healthcare system, future programs can maximize their potential to support healthy aging and intergenerational solidarity.

This study has several limitations. While the research team received qualitative training, some members' relative inexperience in social science methodologies may have influenced the depth of probing and analysis. The sample size was appropriate for qualitative inquiry, but the limited the diversity of participants, particularly from non-Chinese participants, older adults of advanced age, and those who were divorced or widowed, may have constrained the range of perspectives and potentially affected thematic saturation. Although three interviews were conducted by telephone and one via Zoom, the majority were face-to-face, minimizing the potential impact of mode on rapport and depth of responses. Participants may also have responded overly positively about the program knowing some interviewers were involved. Nevertheless, we found most interviewees provided objective views and holistic perspectives of issues discussed. Finally, while this study focused on capturing the breadth of older adults' experiences, it did not compare perceptions by number of YCHV engagements. Such analyses were beyond the scope of this qualitative paper but are being addressed in a separate mixed-methods evaluation of the HealthStart program.

## Conclusion

Engaging YCHVs as health advocates in preventive health programs for older adults is both acceptable and promises health benefits. This study highlights the potential for scaling YCHV programs to benefit wider populations while emphasizing the need for targeted training in behavior change techniques and referral pathways to community resources. Beyond individual health gains, such initiatives hold promise in addressing ageist attitudes, fostering positive intergenerational interactions, and complementing primary care efforts. To maximize impact, future research should explore the long-term effectiveness of YCHV programs and the feasibility of integrating them within preventive healthcare systems.

## Supporting information

**S1 Table. Interview Guide for Older Adult Participants.**
(DOCX)

**S2 Table. COREQ (COnsolidated criteria for REporting Qualitative Research) Checklist.**
(DOCX)

## Acknowledgments

We extend our gratitude to the members of the Singapore General Hospital Division of Population Health and Integrated Care, Youth Corps Singapore, HealthStart participants, and our community partners for their invaluable support. We also thank Dr Sou Kalon for his contributions in refining the manuscript, and Mustafa Chechatwala for submitting the manuscript. The authors also acknowledge the use of ChatGPT (OpenAI) for grammatical checks. The content, interpretations, and conclusions remain the sole responsibility of the authors.

## Author contributions

**Conceptualization:** Audrey Shu Ting Kwan, Nerice Heng Wen Ngiam, Xiaoting Huang, Lynn Pei Zhen Teo, Sheryl Wen Ning Ho, Kennedy Yao Yi Ng, Lian Leng Low.

**Data curation:** Audrey Shu Ting Kwan, Jasmine Yee Ru Cheng, Jed Jasman, Elliott Weizhi Sim, Alicia Shi Yao Chee.

**Formal analysis:** Audrey Shu Ting Kwan, Jasmine Yee Ru Cheng, Jed Jasman, Elliott Weizhi Sim.

**Funding acquisition:** Ka Shing Yow, Jie Xin Lim, Nerice Heng Wen Ngiam, Kennedy Yao Yi Ng, Lian Leng Low.

**Investigation:** Audrey Shu Ting Kwan, Jasmine Yee Ru Cheng, Jed Jasman, Elliott Weizhi Sim.

**Methodology:** Audrey Shu Ting Kwan, Nerice Heng Wen Ngiam, Kennedy Yao Yi Ng, Lian Leng Low.

**Project administration:** Audrey Shu Ting Kwan, Jasmine Yee Ru Cheng, Jed Jasman, Elliott Weizhi Sim, Alicia Shi Yao Chee, Si Qi Lim, Thaddeus Chi En Cheong, Muhammad Razzan Razaki, Xiaoting Huang, Lynn Pei Zhen Teo, Sheryl Wen Ning Ho, Zi Hao Lee, Kharuna Jaichandra.

**Supervision:** Natasha Howard, Kennedy Yao Yi Ng, Lian Leng Low.

**Writing – original draft:** Audrey Shu Ting Kwan, Jasmine Yee Ru Cheng, Jed Jasman, Elliott Weizhi Sim.

**Writing – review & editing:** Audrey Shu Ting Kwan, Jasmine Yee Ru Cheng, Jed Jasman, Elliott Weizhi Sim, Alicia Shi Yao Chee, Ka Shing Yow, Jie Xin Lim, Si Qi Lim, Thaddeus Chi En Cheong, Nerice Heng Wen Ngiam, Muhammad Razzan Razaki, Xiaoting Huang, Lynn Pei Zhen Teo, Sheryl Wen Ning Ho, Zi Hao Lee, Kharuna Jaichandra, Chee Hsiang Liow, Natasha Howard, Kennedy Yao Yi Ng, Lian Leng Low.

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
