## [Decision Letter · Decision Letter 0]

19 Aug 2025

PONE-D-25-28987What do Older Adults Think of Youth Community Health Volunteers (YCHVs)? Perceptions and Experiences of Older Adults in a Youth-Led Post-Health Screening Advocacy Program in Singapore: A Qualitative StudyPLOS ONE?

Dear Dr. Ng,

Thank you for submitting your manuscript to PLOS ONE. After careful consideration, we feel that it has merit but does not fully meet PLOS ONE’s publication criteria as it currently stands. Therefore, we invite you to submit a revised version of the manuscript that addresses the points raised during the review process.

We look forward to receiving your revised manuscript.

Kind regards,

Morufu Olalekan Raimi, Ph.D

Academic Editor

PLOS ONE

Journal Requirements:

This study is supported by the Singapore Ministry of Health’s National Medical Research Council under the Fellowship Programme by SingHealth Regional Health System, Population-based, Unified, Learning System for Enhanced and Sustainable (PULSES) Health Centre Grant (NMRC/CG/C027/2017_SHS), the Healthy, Empowered and Active Living (HEAL) fund and the Infocomm Media Development Authority Digital for Life Fund.

3. In this instance it seems there may be acceptable restrictions in place that prevent the public sharing of your minimal data. However, in line with our goal of ensuring long-term data availability to all interested researchers, PLOS’ Data Policy states that authors cannot be the sole named individuals responsible for ensuring data access (http://journals.plos.org/plosone/s/data-availability#loc-acceptable-data-sharing-methods).

6. Please remove all personal information, ensure that the data shared are in accordance with participant consent, and re-upload a fully anonymized data set.

Additional guidance on preparing raw data for publication can be found in our Data Policy (https://journals.plos.org/plosone/s/data-availability#loc-human-research-participant-data-and-other-sensitive-data) and in the following article: http://www.bmj.com/content/340/bmj.c181.long .

Additional Editor Comments:

Manuscript ID: PONE-D-25-28987

Title: What do Older Adults Think of Youth Community Health Volunteers (YCHVs)? Perceptions and Experiences of Older Adults in a Youth-Led Post-Health Screening Advocacy Program in Singapore: A Qualitative Study

Editorial Decision: Major Revision

Editorial Comments to the Authors

Thank you for submitting your manuscript to PLOS ONE. Both reviewers agree that your study addresses an important and underexplored area in intergenerational and community health, particularly in the context of Singapore’s evolving approaches to chronic disease management. The reviewers find merit in your focus on the perspectives of older adults within a youth-led community health volunteer initiative. However, they have highlighted several areas that require substantive revision before the paper can be considered further.

In particular, reviewers identified the following key issues that must be addressed:

1. Clarity and Completeness of Program Description

Both reviewers noted insufficient detail on the Youth Community Health Volunteer (YCHV) program and the HealthStart intervention. Please expand your description to include:

• Youth CHVs’ age range, recruitment sources, training length, evaluation or competency checks, languages spoken, and whether their participation fulfills academic or community service requirements.

• Basic structure and duration of the HealthStart program (centre-based vs. home-based, individual vs. group delivery).

• Approximate number of YCHVs involved.

This level of detail is essential for readers unfamiliar with the Singaporean context and for ensuring transparency in program description.

2. Methodological Transparency

Several methodological elements require clarification or addition:

• State explicitly the language of interviews (English, or otherwise), including procedures for transcription/translation where relevant.

• Describe interviewer training and background.

• Specify the setting in which interviews were conducted (e.g., quiet room, community centre, home visits).

• Include the sampling method and study setting in the abstract.

• Consider including your interview guide/questions in a supplementary table for transparency.

• Report your study in line with the Consolidated Criteria for Reporting Qualitative Research (COREQ) or similar framework to strengthen methodological rigor.

3. Presentation of Findings

The presentation of qualitative data requires refinement for readability and evidence support:

• Shorten long quotations for easier reading while retaining meaning.

• Where only one quotation supports a point, consider providing an additional supporting quotation to increase evidence robustness.

• Add direct quotations in places where findings are reported in summary form (e.g., “one participant shared…”).

• Insert quotations into long descriptive paragraphs to balance interpretation with participant voices.

• Review the terminology used for presenting themes (e.g., use “revealed” instead of “explored” to better reflect findings).

• Where possible, analyze differences between participants who engaged with YCHVs for fewer vs. more sessions, as this may yield practical insights.

4. Contextualization and Discussion

To aid international readers:

• Provide a brief explanation of Singapore’s healthcare system (e.g., availability of universal health coverage and its implications).

• Clarify Singapore-specific terminology used in participant quotes.

• Avoid repeating themes in the Discussion without additional interpretation.

• Refine the limitations section: your sample size is adequate for qualitative inquiry, but limitations relate more to representation and mode of data collection (e.g., face-to-face vs. non-face-to-face).

• Consider discussing whether a minimum threshold of YCHV engagements is necessary to generate measurable benefit.

5. Language and Style

• Correct specific textual issues noted by Reviewer 1 (e.g., line 184 redundancy, line 224 phrasing, “enlightening” on line 332, and formatting of Conclusion).

• Adjust manuscript title for conciseness as suggested by Reviewer 2: Perceptions and Experiences of Older Adults in a Youth-Led Post-Health Screening Advocacy Program in Singapore: A Qualitative Study.

Decision Summary

Your manuscript makes a valuable contribution, but it currently requires major revision to meet PLOS ONE’s standards for methodological transparency, qualitative rigor, and accessibility to an international readership. Please carefully address all reviewer concerns point by point in your rebuttal letter and highlight revisions in your manuscript.

We look forward to receiving your revised submission.

Sincerely,

Prof. Morufu Olalekan Raimi

Academic Editor

PLOS ONE

Reviewers' comments:

Reviewer's Responses to Questions

**Comments to the Author**

1. Is the manuscript technically sound, and do the data support the conclusions?

Reviewer #1: Yes

Reviewer #2: Yes

2. Has the statistical analysis been performed appropriately and rigorously?

Reviewer #1: Yes

Reviewer #2: Yes

3. Have the authors made all data underlying the findings in their manuscript fully available?

Reviewer #1: Yes

Reviewer #2: No

4. Is the manuscript presented in an intelligible fashion and written in standard English?

Reviewer #1: Yes

Reviewer #2: Yes

Reviewer #1: This is an interesting and important qualitative study of an intergenerational intervention for chronic disease management.

There are a few things that would make the paper stronger:

1. While the authors cite pre-publications detailing the Healthstart program and their mixed methods evaluation, this paper would benefit from more detail on the youth CHV program - specifically: how old are the youth, how long is their training, from where are they recruited, does this provide class credit or community service requirement fulfillment, how many youth CHVs are there, what languages do they speak, etc.

2. Also in the methods, include that interviews were conducted in English, which is one of the official languages of Singapore. Not all readers will be familiar with this.

3. It may also be helpful to the reader to understand whether Singapore has government-sponsored, universal healthcare - - as this is a barrier that some face in other countries (e.g. US).

4. It would also be useful to include the interview questions in a table.

5. Specific line item edits:

Line 184: “such as “reducing reduced their consumption of sugar-sweetened”

Line 224: “to foster of trust and accountability”

Line 277-286 - for the non-resident of Singapore, this quote would benefit from a little more explanation of the terminology used.

Line 332 - perhaps a different word than “enlightening” - - this doesn’t seem to fit.

Line 518: CONCLUSION

Reviewer #2: Review comments:

Thanks for giving me this opportunity to review this manuscript. This study explored older adults’ experiences and perceptions of a YCHV-led health advocacy program following community health screening in Singapore.

Below are my comments for your consideration and improvement:

Title

1. Suggest making the manuscript title more concise: Perceptions and Experiences of Older Adults in a Youth-Led Post-Health Screening Advocacy Program in Singapore: A Qualitative Study

Abstract

2. Please add your sampling method and setting.

3. The first sentence in your conclusion should be shifted to the “results” section.

Introduction

4. To improve the quality of reporting qualitative data, I will suggest that authors consider using “The consolidated criteria for reporting qualitative research”.

Method

Under “study site”,

5. Page 2, please provide some information on the characteristics of the youth volunteers, e.g., age range, training background, and how many days of training they received? any competency check post-training?

6. Page 3, can you provide a short statement on the HealthStart program? how long is the program? is it centre-based? one-to-one or group?

Under “data collection”

7. Page 4, where did you conduct the interviews face-to-face? in a quiet room?

8. What language did you use? Were all interviews conducted in English? If not, please describe how you recorded, transcribed, and translated.

9. Please state the training background of the interviewers.

RESULTS

10. As shown in Table 1, any differences in their perceptions between a smaller number of YCHV engagements and a higher number of YCHV engagements?

11. When presenting your qualitative data, please consider shortening long quotes from just one participant to easy reading, e.g., lines 193-202; 277-286; 384-392;

12. Please consider having at least two quotes (with participants’ codes) will increase the level of evidence in a qualitative study. E.g., line 155, line 312, 340, 378, 410-412;

13. When indicating “one participant shared …”, please add the original quote. E.g., line 161, 268, 272,

14. Please consider adding quotes in your long paragraphs under different sub-themes. E.g., between lines180-190; 208-221;

15. Please change the verb “explored” to “revealed” when presenting themes/sub-themes in your RESULT section. e.g., line 261, 320, 368

DISCUSSION

16. Lines 419-422: You don't have to repeat the 5 themes here.

17. Line 511: As part of your study limitation, your sample size is sufficient for such a qualitative study, rather, the representation of the samples.

18. Under your study limitations, were interviews conducted non-f2f your limitations?

19. It will be good to have a discussion on the differences in perception between those who received 1-3 YCHV sessions and 4-7 sessions, which may inform future practices, like setting a minimum number of YCHV engagement sessions to be more beneficial.

**Do you want your identity to be public for this peer review?** For information about this choice, including consent withdrawal, please see our Privacy Policy

Reviewer #1: **Yes: ** Renee Cadzow

Reviewer #2: **Yes: ** Tianma Xu

---

## [Author Response · Author response to Decision Letter 1]

15 Sep 2025

We thank the editors and reviewers for their thoughtful and constructive comments, which have greatly strengthened the manuscript. We have carefully reviewed each point raised and provide detailed responses below. Revisions have been incorporated into the manuscript as appropriate, with changes highlighted for ease of review.

---

## [Editor Report · Decision Letter 1]

16 Oct 2025

Perceptions and Experiences of Older Adults in a Youth Community Health Volunteer–Led Health Coaching Program in Singapore: A Qualitative Study

PONE-D-25-28987R1

Dear Dear Dr. Ng,,

We’re pleased to inform you that your manuscript has been judged scientifically suitable for publication and will be formally accepted for publication once it meets all outstanding technical requirements.

Kind regards,

Morufu Olalekan Raimi, Ph.D

Academic Editor

PLOS ONE

Additional Editor Comments (optional):

PLOS ONE

Date: 14 October 2025

Dr. Kennedy Yao Yi Ng

Division of Population Health & Integrated Care

Singapore General Hospital

Singapore

Decision on Manuscript PONE-D-25-28987_R1

Perceptions and Experiences of Older Adults in a Youth Community Health Volunteer–Led Health Coaching Program in Singapore: A Qualitative Study

Dear Dr. Ng,

Thank you for submitting your revised manuscript to PLOS ONE and for your thorough and thoughtful responses to the reviewers' comments. I have now carefully assessed the revised version and your point-by-point rebuttal letter.

I am pleased to inform you that your manuscript has been deemed suitable for publication in PLOS ONE, pending minor final formatting checks to ensure full compliance with our journal style.

The revisions you have made have significantly strengthened the paper. You have successfully addressed the core concerns raised during the review process by:

1. The expanded description of the HealthStart program and the Youth Community Health Volunteers (YCHVs), along with the clearer detailing of interview procedures, language translation, and interviewer background, now provides the necessary context and reproducibility for an international readership. The inclusion of the COREQ checklist is a notable strength.

2. The refinement of the results section, through shortened quotes, the addition of supporting quotations, and a better balance between participant voices and researcher interpretation has greatly improved the readability and evidentiary support of your findings.

3. The brief explanation of Singapore's healthcare system and the clarification of local terminology are very helpful. The discussion is now more focused and interpretive, moving beyond a simple restatement of themes to a more meaningful synthesis with the existing literature.

Your manuscript makes a valuable and novel contribution to the literature on intergenerational health initiatives and community-based preventive care for aging populations. The study is well-conducted, the analysis is sound, and the findings offer practical insights for policymakers and practitioners in similar contexts globally.

Congratulations on an excellent piece of work.

Sincerely,

Prof. Morufu Olalekan Raimi, Ph.D.

Deputy Director, Niger Delta Institute for Emerging and Re-emerging Infectious Diseases, Federal University Otuoke, Bayelsa State.
---

## [Editor Report · Acceptance letter]

PONE-D-25-28987R1

PLOS ONE

Dear Dr. Ng,

I'm pleased to inform you that your manuscript has been deemed suitable for publication in PLOS ONE. Congratulations! Your manuscript is now being handed over to our production team.

Kind regards,

on behalf of

Prof Morufu Olalekan Raimi

Academic Editor

PLOS ONE